# Role of C5aR1 and C5L2 Receptors in Ischemia-Reperfusion Injury

**DOI:** 10.3390/jcm10050974

**Published:** 2021-03-02

**Authors:** Carlos Arias-Cabrales, Eva Rodriguez-Garcia, Javier Gimeno, David Benito, María José Pérez-Sáez, Dolores Redondo-Pachón, Anna Buxeda, Carla Burballa, Marta Crespo, Marta Riera, Julio Pascual

**Affiliations:** 1Department of Nephrology, Parc de Salut Mar, 08003 Barcelona, Spain; erodriguezg@psmar.cat (E.R.-G.); mperezsaez@psmar.cat (M.J.P.-S.); mredondopachon@psmar.cat (D.R.-P.); abuxeda@psmar.cat (A.B.); cburballa@psmar.cat (C.B.); mcrespo@psma.cat (M.C.); 2Department of Pathology, Parc de Salut Mar, 08003 Barcelona, Spain; jgimenobeltran@psmar.cat; 3Kidney Research Group, Hospital del Mar Medical Research Institute, IMIM, 08003 Barcelona, Spain; dbenito@psmar.cat

**Keywords:** kidney transplant, ischemia-reperfusion injury, delayed graft function, complement system, C5a receptors

## Abstract

The role of C5a receptors (C5aR1 and C5L2) in renal ischemia-reperfusion injury (IRI) is uncertain. We generated an in vitro model of hypoxia/reoxygenation with human proximal tubule epithelial cells to mimic some IRI events. C5aR1, membrane attack complex (MAC) and factor H (FH) deposits were evaluated with immunofluorescence. Quantitative polymerase chain reaction evaluated the expression of *C5aR1*, *C5L2* genes as well as genes related to tubular injury, inflammation, and profibrotic pathways. Additionally, C5aR1 and C5L2 deposits were evaluated in kidney graft biopsies (KB) from transplant patients with delayed graft function (DGF, *n* = 12) and compared with a control group (*n* = 8). We observed higher immunofluorescence expression of C5aR1, MAC and FH as higher expression of genes related to tubular injury, inflammatory and profibrotic pathways and of *C5aR1* in the hypoxic cells; whereas, *C5L2* gene expression was unaffected by the hypoxic stimulus. Regarding KB, C5aR1 was detected in the apical and basal membrane of tubular epithelial cells, whereas C5L2 deposits were observed in endothelial cells of peritubular capillaries (PTC). DGF-KB showed more frequently diffuse C5aR1 staining and C5L2 compared to controls. In conclusion, C5aR1 expression is increased by hypoxia and IRI, both in vitro and in human biopsies with an acute injury. C5L2 expression in PTC could be related to endothelial cell damage during IRI.

## 1. Introduction

Organ injury because of ischemia followed by reperfusion is a major clinical problem. Renal ischemia-reperfusion injury (IRI) is the most common cause of acute kidney injury after renal transplantation [1]. The pathophysiology of IRI is characterized by three different processes: neutrophilic infiltration, reactive oxygen species formation, and innate immune activation [2,3,4,5]. Regarding innate immune activation in renal IRI, complement activation plays a critical role [6,7]. It may occur through three different pathways (classic, alternative, or lectin) due to the interaction with antibodies, pro-inflammatory proteins, or mannose-binding lectin fragments [8,9,10]. Regardless of the mechanism, complement activation leads to the generation of the terminal panel of complement anaphylatoxins C3a, C5a, and C5b-9-membrane attack complex (MAC) that can enhance the recruitment of inflammatory cells and intensify cellular lysis within the site of tissue/organ injury. MAC and FH’s role in IRI after kidney transplant (KT) has been described in previous studies [11,12,13,14,15]. However, the role of C5a receptors during IRI in human tissues is less known. C5a is one of the most potent inflammatory mediators, which induces leukocyte chemotaxis, activates leukocytes and endothelial cells, and drives the production of inflammatory mediators. C5a binds to two receptors, named C5aR1 and C5aR2 (also known as C5L2) [16]. Several studies reported the critical role of the C5a/C5aR1 interaction-mediated inflammation in the pathogenesis of renal injury [17,18,19].

Nevertheless, the function of C5a/C5L2 interactions in the inflammatory process and its involvement in pathophysiology is controversial. Although opposite anti-inflammatory and pro-inflammatory effects have been reported, this seems to rely on different methodologies, such as disease models and cell types [20]. *C5aR1*- and *C5L2*-knockout mice have shown a protective effect from IRI, with less fibrosis and better tubular regeneration [21,22]. Up to date, only a previous study has addressed the evaluation of C5aR1 and C5L2 expression in KT patients without conclusive results. In a small number of patients, Van Werkhoven et al. were unable to find differences between C5aR1 or C5L2 expression in kidney biopsies obtained pre-donation from kidney living donors and post-transplant kidney biopsies showing acute tubular necrosis [23].

In recent years, C5a-receptor inhibitor drugs have been developed and are being used in other kidney pathologies in which complement also participates [24]. Expanding the knowledge about the role of C5a receptors in IRI in kidney transplants is essential to consider the usefulness of these drugs in this setting.

Our study aimed to evaluate the role of C5aR1 and C5L2 during IRI in an in vitro human cell model of hypoxia/reoxygenation and post-transplant kidney biopsies with ischemic injury.

## 2. Methods

### 2.1. Cell Culture

HK-2, an immortalized proximal tubular epithelial cell (PTEC) line derived from normal kidneys, was employed for in vitro studies The HK-2 cell line was obtained from Dr. López-Novoa. Cells were cultured in RPMI 1640 supplemented with 10% fetal bovine serum (FBS), 1% P/S, 1% glutamine and 1% ITS (insulin–transferrin–selenium) under normal conditions (air O_2_ and 5% CO_2_ at 37 °C). Products were purchased from Biowest (Nuaillé, France) and Life Technologies (Waltham, MA, USA). Subconfluent cells were cultured for 48 h in normal conditions and, after changing cell media, for 48 h more either under normal conditions or under hypoxic conditions (1% O_2_ and 5% CO_2_ at 37 °C), based on the previous model [25]. After this time, the cell medium was replaced and incubated for 30 min under normal conditions. Throughout the reoxygenation period, the cell medium contained 10% heat-inactivated FBS or 10% human serum (NHS, Sigma-Aldrich, St. Louis, MO, USA), according to the study group. As a result, the experimental conditions were as follows: (i) normoxia + 10% FBS (N+FBS), (ii) normoxia + 10% NHS (N+NHS), (iii) hypoxia + 10% FBS (H+FBS) and (iv) hypoxia + 10% NHS (H+NHS).

For immunofluorescent protein detection, the same experiments were performed on sterilized glass coverslips on 6-well plates. After incubations, cells were washed with PBS, fixed in 2% PFA for 15 min and permeabilized at −20 °C in methanol for 5 min more. Primary antibodies against C5b-9 (#HM2167, clone 6G3, Hycult Biotech, Uden, The Netherlands) (mAb 1:2000), factor H (clone 214, courtesy of Dr S Rodríguez de Córdoba, CSIC) (mAb 1:2000), C5aR (#HM2094, clone S5/1, Hycult Biotech) (mAb 1:50) and β-actin (#A1978, Sigma-Aldrich) (mAb 1:6000) were diluted in 1.5% BSA and incubated at room temperature for 1 h. Secondary antibodies coupled to AlexaFluor^®^ 489 or 549 (Life Technologies, Waltham, MA, USA) (1:2000) were used to localize proteins. Coverslips were mounted on glass slides using Mowiol/DABCO+DAPI solution (all from Sigma-Aldrich, St. Louis, MO, USA).

Cell staining was evaluated with an upright Nikon Ni-e microscope (Amsterdam, Netherlands). The intensity of the staining was measured with Image J software v1.51j8 (Bethesda, MD, USA), and results were expressed as arbitrary units of fluorescence (AUF) per cell after counting the number of nuclei per area, as previously published [26,27].

For gene expression, cells grown in monolayer after the reoxygenation period were collected with cell scrapers and washed in PBS at 300 g for 5 min. Pellets were kept frozen at −80 °C in TriPure isolation reagent (Roche, Manheim, Germany) until total RNA extraction. RNA quantity and purity were analyzed with NanoDrop (ND-1000 V3.3, Waltham, MA, USA). First-strand cDNA was synthesized from 0.5 μg of RNA using the high-capacity cDNA reverse transcription Kit (Applied Biosystems, Foster City, CA, USA) incubated 10 min at 25 °C, 120 min at 37 °C and 5 min at 85 °C. Real-time PCR was performed in the LightCycler^®^ 480 System (Roche, Manheim, Germany) using SYBR Green I Master mix and the ΔΔCp method for relative quantification was used with *PPIA* gene normalization. The sequences of primers were the following:

*C5aR1*: 5′-CACCAAGACACTCAAGGTGGT-3′; 5′-ATTATCCCCGTCACCTGGTAG-3′

*C5L2*: 5′-ATGGAGGTGTAGGCTGGAGAG-3′; 5′-CTATGCCTGAAGCCAGTCTTG-3′

*Col1A1*: 5′-AGTACTGGATTGACCCCAACC-3′; 5′-GTACACGCAGGTCTCACCAGT-3′

*Vimentin*: 5′-TCAATGTTAAGATGGCCCTTG-3′; 5′-GCAGAGAAATCCTGCTCTCCT-3′

*CD59*: 5′-GCCTTCATCCCTAAGTCAACAC-3′; 5′-TTAGAATGTGGCAGCAAGAGAA-3′

*Kim1*: 5′-GCTTTGCAAAATGCAGTTGA-3′; 5′- GGGTCTTAGTCCGTGGCATA-3′

*PPIA*: 5′-GACCCAACACAAATGGTTCC-3′; 5′-TTTCACTTTGCCAAACACCA-3′

### 2.2. Kidney Biopsies

We evaluated the C5aR1 and C5L2 deposits in kidney biopsies from KT patients with delayed graft function (DGF), defined as dialysis needed during the first week after KT, and evidence of acute tubular damage in the biopsy (*n* = 12). Eight patients with one-year protocol graft biopsies with stable creatinine serum values since transplantation, without proteinuria, anti-HLA antibodies, or histological signs of renal dysfunction or rejection served as controls.

The mean time between KT and kidney biopsy for DGF was 15 days (IQR 10.7–29). Biopsies from controls were performed 12 months (IQR 12–16) after KT.

#### Immunohistochemistry

Frozen samples in OCT (Optimal Cutting Temperature compound) stored at −80 °C were available for the study. 6 µm cryostatic sections were mounted onto Superfrost^®^ glass slides (Waltham, MA, USA). After drying at room temperature, we fixed the samples in 10% buffered formalin for 10 min. After PBS washing, 3% goat serum in PBS was used to block nonspecific binding sites for 1 h. Primary antibodies against C5aR1 (#HM2094, Hycult Biotech) (mAb 1:10) and C5aL (#HP9036, Hycult Biotech) (pAb 1:25) were then incubated for one hour more. Blocking solution was employed as diluent. Secondary antibodies conjugated with HRP-labeled polymer (Dako, Glostrup, Denmark) were incubated for 1 h. After this, we visualized the proteins with The EnVision^®^ System-HRP (Dako, Glostrup, Denmark). Sections were counterstained with hematoxylin, dehydrated in graded alcohols, and mounted with DPX (Dibutylphthalate Polystyrene Xylene) mounting media. Samples were histologically evaluated by two independent observers (CA-C and JG) in a blinded manner using a semiquantitative method based on the percentage of stained tubules or capillaries. This method was used and validated in previous studies from our group [13]. Regarding C5aR1 staining, we considered “minimally or focal positive staining” as positive staining in less than 50% of all tubules from the slide and “diffuse staining” as positive staining in more than 50% of tubules.

Regarding C5L2 staining, we applied the same method for C4d staining according to the Banff group criteria [28]. This means “minimally positive” when positive staining in less than 10% of all peritubular capillaries, “focally positive staining” when positive staining in 10–50% of all PTC, and finally “diffusely positive” for positive staining in more than 50% of all PTC.

The intensity of staining was graded as 0 (no staining), +1 (staining visible at 40× magnification), +2 (at 20×), +3 (at 10×), and finally +4 (at 2–4×). We considered “high-intensity” as staining visible at or below 10× magnification (+3 or +4).

### 2.3. Statistical Analysis

We reported continuous variables with a normal distribution as mean and standard deviation; if variable distribution deviated significantly from normality, we reported median and interquartile range (IQR).

We used the Mann–Whitney U test to compare the means of continuous variables and the chi-squared test to analyze categorical data. A *p*-value (two-tail) < 0.05 was considered statistically significant.

All analyses were performed using SPSS V 25.0 (SPSS Inc., Chicago, IL, USA).

## 3. Results

### 3.1. Expression of MAC and FH-Related Proteins in HK-2 Cells

To establish an in vitro model for hypoxia/reoxygenation injury on PTEC, we cultured HK-2 cells under normoxic or hypoxic conditions. Data from five different experiments with the same conditions were analyzed.

The highest mean intensity value for MAC expression was observed in H+NHS cells, followed by H+FBS cells. We did not find significant differences between these two groups (*p* = 0.588). H+NHS cells showed higher MAC intensity values when compared with N+FBS cells (*p* = 0.005) and with N+NHS cells (*p* = 0.030). We did not find statistical differences between H+FBS cells and N+NHS cells. H+FBS cells showed higher values than N+FBS cells (*p* = 0.021). Finally, we detected a significantly higher MAC expression in N+NHS cells compared with N+ FBS cells (*p* = 0.002; Figure 1).

Regarding FH, similarly to MAC, the highest mean intensity values were observed in H+NHS cells, followed by H+FBS cells. However, H+NHS cells showed higher values when compared with H+FBS cells (*p* < 0.023). The FH mean intensity values in H+NHS cells and H+FBS cells were higher than all the other experimental conditions. In normoxic cells, FH mean intensity values were higher in the N+NHS cells compared with N+FBS cells (*p* = 0.037; Figure 1).

### 3.2. Gene Expression in HK-2 Cells

The *C5aR1* gene expression was higher in hypoxic cells than in normoxic (*p* = 0.001 vs. N+FBS; *p* = 0.005 vs. N+NHS cells). Cells exposed to hypoxia+NHS showed the highest gene expression values. However, they were similar to the values found for H+FBS cells (*p* = 0.155), indicating a strong influence of the oxygen deprivation period. Regarding C5L2, only cells exposed to 10% NHS increased the expression of this gene, independently to normoxic or hypoxic conditions (Figure 2).

Gene expression for complement regulator CD59 was higher in hypoxic cells when compared with normoxic cells, independently to 10% NHS addition (Figure 2).

Kidney injury molecular 1 (*KIM-1*) gene expression was also higher in hypoxic cells when compared with N+FBS cells (*p* = 0.016 to H+NHS and *p* = 0.006 to H+FBS) (Figure 2).

Pro-fibrotic, Collagen A-1 and vimentin gene expression were increased in both H+NHS and H+FBS cells when compared with N+FBS. No differences were found between normoxic nor hypoxic cells, independently of the addition to NHS (Figure 2).

### 3.3. Expression of C5aR1 Protein in HK-2 Cells

We assayed C5aR1 protein localization by immunofluorescence stain in our hypoxia/reoxygenation model. We analyzed the data from three different experiments in the same condition described above. We detected a significant increase of C5aR1 expression in the H+NHS cells compared with N+FBS cells (*p* = 0.049). The N+NHS and H+FBS cells also tended to express more C5aR1 protein than the N+FBS cells. However, these differences were not statistically significant (Figure 3).

### 3.4. Immunohistochemistry for C5aR1 andC5L2 in Kidney Tissue Samples

We studied twelve kidney biopsies from patients with DGF and acute tubular damage and eight kidney biopsies from controls. The global sample’s median age was 61.0 ± 10.5 years; they were predominantly males (75% vs. 16.7%). Basal characteristics were similar between DGF and the control group (Table 1). We found a higher serum creatinine value at 12 months after transplantation in the group with DGF compared with controls (2.16 mg/dL vs. 1.14 mg/dL; *p* = 0.004). None of these patients had a prior history of eventual glomerulopathies related to complement alterations.

We detected C5aR1 deposits in either the apical or the basal regions of the tubular epithelial cell membrane. No lateral membranous nor cytoplasmic staining was detected in tubular cells. We did not observe positive staining of other renal structures such as glomeruli or vessels.

Renal biopsies from patients experiencing DGF showed more frequently diffuse-positive staining (44% vs. 13%; *p* = 0.048) of both proximal and distal tubules, whereas C5aR1 staining in sample controls was minimally or focally positive and was located predominantly in distal tubules (Figure 4).

We observed C5L2 staining in the endothelium of peritubular capillaries. Diffusely positive staining was only observed in samples from patients experiencing DGF (67% vs. 0%; <0.001; Figure 4).

## 4. Discussion

In the present study, renal proximal tubular epithelial cells (PTEC) in culture show that hypoxia/reoxygenation activates the complement system, as reflected by protein expression of the final complement pathway’s effector MAC and the relevant complement regulator FH. Moreover, we find an increased gene and protein expression of *C5aR1* in hypoxic cells and diffuse and intense deposits of C5aR1 in tubular epithelial cells from KT biopsies of patients with DGF. Our results show that C5L2 expression appeared not to be related to hypoxia and only observed in the endothelium of peritubular capillaries but not in tubular epithelial cells. To our knowledge, this is the first study evaluating jointly the gene expression of *C5aR1* and *C5L2* in PTEC undergoing hypoxia/reperfusion and their protein expression in KT biopsies with acute tubular damage.

IRI is one of the most important causes of DGF after kidney transplantation (KT) [29], a form of acute tubular injury that negatively impacts graft outcomes [29,30,31,32]. Renal proximal tubular epithelial cells located at the corticomedullary zone are relatively susceptible to IRI and identified as one of the main targets of complement activation [25]. The main pathway responsible for complement activation during IRI in humans remains unclear [8]. For this reason, we focused the study on final complement fractions, MAC, and C5a to delineate the process of their activation in the context of renal ischemia/reperfusion injury.

We detected increased MAC and FH protein deposits by immunofluorescence in PTEC after hypoxic stress, independently of human serum addition. These results suggest a local production of MAC and FH from HK-2 cells during hypoxia/reoxygenation in our model. This finding is in concordance with previous studies using this cell line, reporting the production of different complement components, including MAC and FH. We are the first to report the expression of C5a receptors in an experimental model using HK-2 cells. Although previous works described the role of MAC and FH in IRI, they mostly were on animal models [15,17,33,34]. Our group formerly reported a relationship between increased plasmatic MAC levels and histological MAC and FH deposits and the severity of acute kidney injury in either native and transplanted kidney [13]. However, the relationship with C5a receptors at that time was not studied. In the present study, we detect the increased expression of the C5aR1 in tubular cells under hypoxia/reoxygenation conditions. These experimental data are confirmed when we detect C5aR1 deposits in tubular epithelial cells from kidney biopsies. C5aR1 deposits are observed both in controls and in biopsies of patients who underwent DGF; however, these deposits are expressed more diffusely, affecting both proximal and distal renal tubules in those samples with evidence of acute renal damage, which could reflect a greater activation of the complement in these patients. Contrary to *C5aR1*, the gene expression of *C5L2* in tubular epithelial cells appears to be only regulated by the presence of human serum. We could not find any influence of hypoxic stimuli, neither with the expression of pro-inflammatory nor with the expression of profibrotic genes in our in vitro model. Other investigators evaluated renal C5aR1 under diverse inflammatory conditions, and they reported its expression in both proximal and distal tubule [35,36]. In immune cells, C5L2 is expressed alongside C5aR1, and it is thought to serve as a decoy receptor for C5a, thereby counterbalancing C5aR1-initiated responses [37,38]. Both C5aR1 and C5L2 expression have been found in non-immune tissues as well, such as tubular cells [39]. Thus, several studies based on murine models reported protection against IRI, with less inflammatory infiltration and fibrosis when the expression of C5aR1 or C5L2 [21,22] was knockdown. As we mentioned above, the role of C5a receptors in IRI is described in several animal models. However, there is evidence indicating relevant differences between the complement system’s functioning between mice and humans. On one hand, murine models reflect that activation of the complement system during IRI occurs almost exclusively through the alternative pathway with the lectin pathway’s possible participation. These results are not clearly in studies carried out in humans, in which even the participation of the classical pathway is described [25]. Furthermore, human studies with complement inhibitors, based on murine models, have not shown goods results in terms of protection against IRI damage [8]. For these reasons, extrapolating the results from murine models to humans should be done with great caution. These discrepancies and the growing number of complement inhibitor drugs currently developing [40] make it essential to know the complement components involved in ischemia-reperfusion damage in humans.

Prior to our work, only one previous study evaluated the role of C5a receptors in kidney transplant biopsies. Van Werkhoven et al. [23] assessed the expression of renal C5aR1 and C5L2 in biopsies from living donors and patients suffering from acute tubular necrosis (ATN), acute cellular and vascular rejection and interstitial fibrosis and tubular atrophy (IF/TA). The authors reported C5aR1 and C5L2 expression predominantly in the distal tubule at the epithelial cell basal membrane but in different regions. C5aR1 was mainly expressed in the thick ascending limb of Henle’s loop, and C5L2 expression was predominantly in the distal convoluted tubule. Double staining of C5aR and C5L2 revealed the expression of these two receptors to be almost mutually exclusive. In this work, the intensity of renal C5aR1 expression was comparable between normal biopsies obtained from living donors and post-transplant biopsies (ATN, acute rejection, and IF/TA). However, in Van Werkhoven’s study, the extension of the staining was not evaluated. We do not detect apparent methodological differences that may explain the different histological patterns reported in their study compared with our results. These differences could be related to different clinical profiles between the two cohorts. Unfortunately, the lack of sufficient clinical information does not allow us to make this comparison.

Nevertheless, ischemia does not exclusively lead to alterations of epithelial cells, also causing interstitial inflammation and interstitial microvasculopathy [41]. Even though renal hypoperfusion mainly induces functional and structural alterations of the tubular epithelium, some studies pointed towards the role of postischemic endothelial cell dysfunction (ED) in peritubular capillaries as a crucial perpetuating factor of prolonged kidney malfunction [42,43]. This microvasculopathy is characterized by cell swelling, increased paracellular and transcellular endothelial permeability, and increased endothelial expression of different cell adhesion molecules. Among those molecules are P-selectin, E-selectin, and ICAM-1, which mediate leukocyte-endothelial interactions, a prerequisite for transvascular leukocyte migration. In this line, an experimental study performed by Thorenz et al. [22] reported immune cells expressing both receptors C5aR1 and C5L2 in the perivascular space of large vessels, and receptor-positive immune cells surrounded the vessels and tubules after IRI. Furthermore, the peritubular capillary network’s integrity was significantly better in *C5aR2* knockout mice after IRI, indicating a role of C5a receptors in the vasculopathy induced after ischemic injury

For these reasons, we believe that C5L2 deposits in peritubular capillaries could be a marker of interstitial microvasculopathy due to DGF. To our knowledge, this is the first study that describes the presence of C5L2 deposits in peritubular capillaries of kidney biopsies from patients experiencing DGF.

Our work has some limitations. First, the lack of specific functional studies does not allow us to establish a direct cause–effect mechanism between complement activation and stimulation of inflammatory or profibrotic pathways. However, we firmly believe that the design of our hypoxia-reoxygenation model covering different experimental situations shows an evident relationship. On the other hand, the limited number of kidney biopsies makes it challenging to explore the relationship between different histological patterns and other predictive variables like cold ischemic time, graft function, or transplant survival.

In conclusion, our hypoxia/reoxygenation injury model proves to be reliable to evaluate the role of the complement system during this process. We report relevant data regarding the role of C5aR1 and C5L2 during IRI in humans, information underrepresented in the literature available to date, which is based mainly on murine models. We found a clear relationship between IRI and a higher expression of both MAC and C5aR1. The higher expression of MAC and C5aR1 was related to complement regulators’ activation, pro-inflammatory and profibrotic markers. These findings are in concordance with the presence of extensive deposits of C5aR1 in the tubular cell membrane of biopsies with acute tubular injury, which is nothing else than the clinical manifestation of IRI. We do not detect a direct implication of C5L2 in tubular damage during IRI; however, it could reflect endothelial activation in patients with DGF. Despite this exciting finding, additional studies, including hypoxic endothelial models, are necessary to confirm this hypothesis.

## Figures and Tables

**Figure 1 jcm-10-00974-f001:**
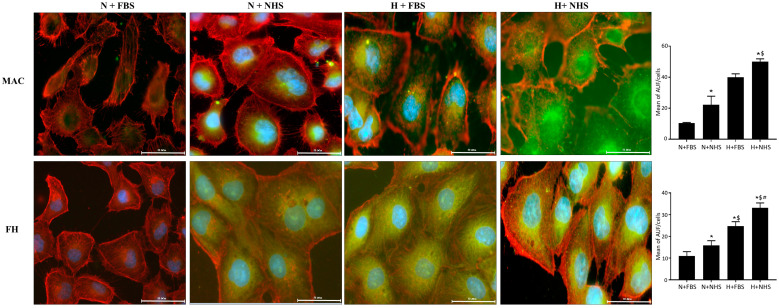
Cellular distribution of membrane attack complex (MAC) and factor H (FH). HK-2 under normoxic or hypoxic conditions for 48 h and later reoxygenated for 30 min with 10% heat-inactivated fetal bovine serum (FBS) or 10% normal human serum (NHS). Two-color immunostaining was performed with MAC (green) and β-actin (red) and with FH (green) and β-actin (red), respectively. Nuclei were visualized by DAPI (blue) (scale bar: 50 μm). The intensity of the staining was expressed as arbitrary units of fluorescence (AUF) per cell. All analyses were performed in a blinded fashion. Data come from 5 different experiments. Data were shown as mean ± SEM and analyzed by the Mann–Whitney test. The experimental conditions were: normoxia + 10% FBS (N+FBS), normoxia + 10% NHS (N+NHS), hypoxia + 10% FBS (H+FBS) and hypoxia + 10% NHS (H+NHS). * *p* < 0.05 vs. normoxia + 10% FBS condition. $ *p* < 0.05 vs. N+NHS, # *p* < 0.05 vs. H+FBS. N: normoxia, H: hypoxia.

**Figure 2 jcm-10-00974-f002:**
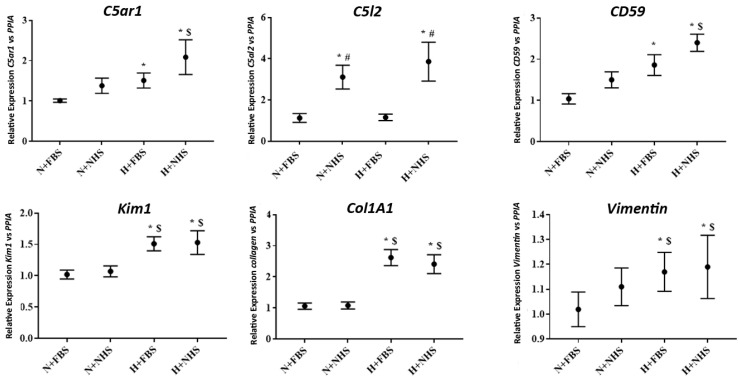
Gene expression of *C5aR1*, *C5L2*, complement regulators, an inflammatory and profibrotic gene from in vitro ischemia-reperfusion injury (IRI) model. Quantitative real-time RT–PCR for analyzing *C5aR1*, *C5L2*, *CD59*, *KIM-1*, *Colagen1A1*, and *Vimentin* was performed in HK-2 cells collected after a reoxygenation step from the IRI model. Gene expression was expressed as relative gene expression vs. *PPIA*. Data were shown as mean ± SEM and analyzed by the Mann–Whitney test. * *p* < 0.05 vs. normoxia + 10% FBS condition. $ *p* < 0.05 vs. N+NHS, # *p* < 0.05 vs. H+FBS.

**Figure 3 jcm-10-00974-f003:**
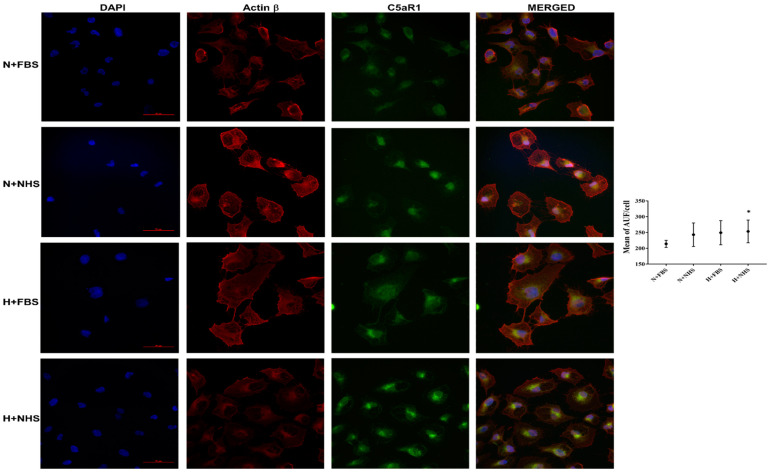
Cellular distribution of C5aR1. HK-2 under normoxic or hypoxic conditions for 48 h and later reoxygenated for 30 min with 10% heat-inactivated fetal bovine serum (FBS) or 10% normal human serum (NHS). Double immunostaining was performed with C5aR1 (green) and β-actin (red). Nuclei were visualized by DAPI (blue) (scale bar: 50 μm). The intensity of the staining was expressed as arbitrary units of fluorescence (AUF) per cell. Data from 3 different experiments are shown as mean ± SEM and analyzed by the Mann–Whitney test. The experimental conditions were: normoxia + 10% FBS (N+FBS), normoxia + 10% NHS (N+NHS), hypoxia + 10% FBS (H+FBS) and hypoxia + 10% NHS (H+NHS). * *p* < 0.05 vs. normoxia + 10% FBS condition.

**Figure 4 jcm-10-00974-f004:**
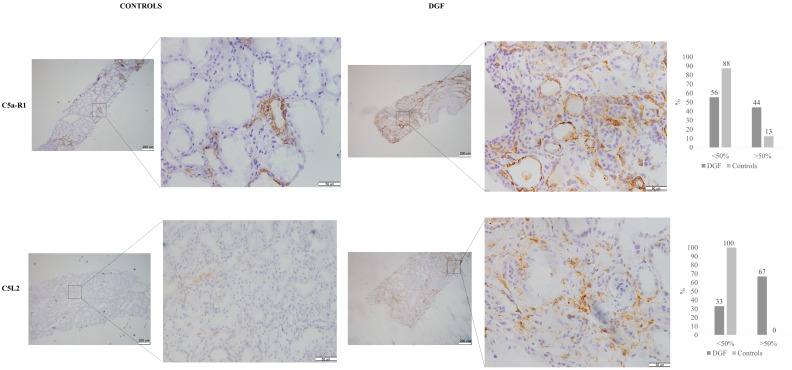
C5aR1 and C5L2 stain in biopsies from kidney transplantation (KT) patients with DGF and one-year protocol biopsies without tissue damage as a control group. C5aR1 and C5l2 were evaluated in the tubular compartment using immunohistochemistry and semiquantitative scoring. C5aR1 stain was observed in the apical and basal side of the membrane from tubular epithelial cells, while C5L2 deposit was observed in the endothelium from peritubular capillaries. DGF biopsies showed more frequently diffuse staining (>50% of tubules or PTC) for both C5aR1 and C5aL2 than controls.

**Table 1 jcm-10-00974-t001:** Baseline characteristics of DGF and controls patients.

	Controls (*n* = 8)	DGF (*n* = 12)	*p*-Value
Age at transplantation (y, mean ± SD)	55.1 ± 9.97	60.9 ± 10.23	0.219
Donor age	56.0 ± 9.81	64.3 ± 11.55	0.185
Recipient sex male (*n*, %)	4 (50)	11 (92)	0.088
History of DM (*n*, %)	2 (25)	6 (50)	0.160
History of hypertension (*n*, %)	7 (87.5)	12 (100)	0.191
Pretransplant hemodialysis (*n*, %)	4 (50)	10 (83)	0.116
Donor Type (*n*, %)			0.260
DBD	6 (75)	6 (50)	
DCD	2 (25)	6 (50)	
12-month creatinine mg/dL (mean ± SD)	1.14 ± 0.27	2.16 ± 1.11	0.004
Cold ischemic time (hours, mean ± SD)	10.6 ± 6.86	15.3 ± 6.81	0.076

DGF, delayed graft function; SD, standard deviation; DM, diabetes mellitus; DBD, donor after brain death; DCD, donor after cardiac death.

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
