# Peer review of "Role of C5aR1 and C5L2 Receptors in Ischemia-Reperfusion Injury"

_jcm, 2021, doi:10.3390/jcm10050974_

Round 1
Reviewer 1 Report
Authors did not add any experiment to their paper and add the limitation of the protein expression analysis and the lack of endothelial cell evaluation.
To enhance the soundness of their work, many experiments should be added. With these limitations, they focus on a precise pathological mechanism and describe it quite well.
Line 256: the term acute kidney injury refers to a specific cilnico-pathological feature in nephrology. Referring to DGF as acute kidney injury is quite inappropriate, authors should change the term and adequate accordingly in the following text.
Author Response
Authors did not add any experiment to their paper and add the limitation of the protein expression analysis and the lack of endothelial cell evaluation.
R// Thank you for taking the time to review our article.
In comparison with our previously submitted paper, in this new version, we added the protein expression of C5aR1 by immunofluorescence assays; we reflect these results in figure 3. So, the phrase about our previous limitation in detecting protein expression was removed from the original manuscript. We agree with the limitation of endothelial cell studies, and we take into account these comments for future works. Our main objective in this study was the renal tubular cells and we did not anticipate any result relating endothelial cells.
To enhance the soundness of their work, many experiments should be added. With these limitations, they focus on a precise pathological mechanism and describe it quite well.
R// We appreciate the comment.
As we described in our conclusion, we believe that our experimental model allowed us to evaluate the role of C5a receptors during the hypoxia-reoxygenation injury reliably.
Line 256: the term acute kidney injury refers to a specific cilnico-pathological feature in nephrology. Referring to DGF as acute kidney injury is quite inappropriate, authors should change the term and adequate accordingly in the following text.
R// We appreciate the advice.
We fully agree that acute kidney injury and DGF could not be considered synonyms. We wanted to point out that we evaluated the role of C5a receptors in patients with acute damage secondary to ischemia-reperfusion injury; probably the correct form to reflect that is to change the term acute kidney injury by acute tubular damage.
Reviewer 2 Report
Thank you for submitting good manuscript to our journal. It is well designed, valuable study with clear conclusion. However, we ask a couple of comment to researchers for the readers of JCM.
First, did you compare recipients’ renal function or cold ischemic time by C5a receptors expression (12 patients)? If you look intensity of IHC and patients’ renal function together, it would be helpful that understand inflammation and tubular (or interstitial) C5aR changes.
Second, biopsy results of the control patients are those of 1 yr after transplantation. We’re afraid that immediate post-Op biopsy might show larger amount of C5aR expression after operation related mild ischemic insult. It would be an another limitation of this study.
Bring together, your figure 4 control showed mild uptake even after stabilized renal condition (normal kidney function?), was it related with CNI effect or other ischemia? Does normal renal tubule show this amount of intensity? Need further explanation. It is better to add patients’ information (average serum creatinine of 8 controls, age, diabetes……) and compare ischemia vs control group (using table, 12 vs 8 ) for the JCM readers’ comprehension.
Thank you.
Author Response
Thank you for submitting good manuscript to our journal. It is well designed, valuable study with clear conclusion. However, we ask a couple of comment to researchers for the readers of JCM.
First, did you compare recipients’ renal function or cold ischemic time by C5a receptors expression (12 patients)? If you look intensity of IHC and patients’ renal function together, it would be helpful that understand inflammation and tubular (or interstitial) C5aR changes.
R// We are very grateful for your review.
We agree that this analysis could improve the explanation of our results. In this line, we compared mean values of serum creatinine levels at 12 months after transplantation between patients showing diffuse stain (>50%) and focal stain (<50%) for both receptors. We found a tendency to higher creatinine values in these patients with C5aR1 diffuse stain than those with focal stain (2.4 ± 1.35 mg/dl vs. 1.7 ± 0.20 mg/dl); however, this difference was not statistically significant (p=0.212); we obtained similar results with the C5L2 stain (2.09 ± 0.90 mg/dl vs. 2.25 ± 1.43; p=0.831). Otherwise, we found no correlation between cold ischemic time and the intensity or extension of the staining. However, we consider these results difficult to interpret in such a small sample, so we decided not to add them to our study results. We reflected this limitation in our manuscript (lines 340-342).
Second, biopsy results of the control patients are those of 1 yr after transplantation. We’re afraid that immediate post-Op biopsy might show larger amount of C5aR expression after operation related mild ischemic insult. It would be an another limitation of this study.
R// We fully agree that this could be a limitation in our study. Unfortunately, we do not perform perioperative biopsies in our center, to confirm or discard this point. However, previous studies including one from our group (Arias-Cabrales, C.E et al. Clinical Kidney Journal Oct-2020), described an increase in complement proteins predominantly in patients who developed DGF, this suggests that complement activation in cases of mild ischemia could be less relevant.
Bring together, your figure 4 control showed mild uptake even after stabilized renal condition (normal kidney function?), was it related with CNI effect or other ischemia? Does normal renal tubule show this amount of intensity? Need further explanation.
R// We appreciate the comment. Indeed, the controls' biopsies were not wholly negative, especially in the case of staining for C5aR1. This finding is because the expression of C5a receptors in the distal renal tubule can be detected under normal conditions; however, in our study, we detected that this expression is more diffuse in DGF biopsies, also affecting proximal tubules in the case of C5aR1 and peritubular capillaries in the case of C5L2. We explain this point in the discussion section (lines 303 to 317).
It is better to add patients’ information (average serum creatinine of 8 controls, age, diabetes……) and compare ischemia vs control group (using table, 12 vs 8 ) for the JCM readers’ comprehension.
R// We welcome the suggestion, and we added this comparison in the results section (lines 219-227)
This manuscript is a resubmission of an earlier submission. The following is a list of the peer review reports and author responses from that submission.
Round 1
Reviewer 1 Report
Arias Cabrales et al describe the potential role of C5a receptors in the field of IRI in kidney transplant. A field in which the role of complement is becoming more and more studied due to the availability of specific therapies.
the experimental design is correct and depicts the possible role of the two different receptors. Did the authors perform immunofluorescence evaluation of C5aR1 and C5aL2? It would be useful to demonstrate the protein expression of at least C5aR1 with Western Blot. Maybe they have negative results from IF.
As authors already comment, one limit of their study is the lack of the analysis in endothelial cells. This evalutation could, of course help in the explanation of their results about the expression of C5aL2 in peritubular capillaries and improve the soundness of their work.
Moreover, the evaluation could be completed if authors could obtain an analysis of pre-implantation biopsies of the same patients already analysed, if possibile.
Author Response
Did the authors perform immunofluorescence evaluation of C5aR1 and C5aL2? It would be useful to demonstrate the protein expression of at least C5aR1 with Western Blot. Maybe they have negative results from IF.
Antibodies used for C5aR1 and C5aL2 detection on renal biopsies were purchased from HycultBiotech (HM2094 and HP9036, respectively). The manufacturer recommends these antibodies for immunohistology applications on frozen sections as we performed. However, only the C5aL2 antibody is suitable for immunofluorescence detection. This technical issue would partially contribute to the message in the manuscript, so we decided not to test on tubular cell culture.
Regarding western blot detection, we thank the reviewer for this suggestion. Unfortunately, we did not perform this test in our in vitro model before, so we need several weeks to obtain cell lysate protein to test the expression of C5aR1 and C5aL2 by western blot technique. We added a phrase in the discussion section to explain this as a limitation (lines 282-283)
As authors already comment, one limit of their study is the lack of the analysis in endothelial cells. This evalutation could, of course help in the explanation of their results about the expression of C5aL2 in peritubular capillaries and improve the soundness of their work.
We thank the reviewer for his/her comment. We were mainly interested in renal tubular cells, and the focus on endothelial cells appeared later with the results. Sure, this study will be part of the next projects on this subject.
Moreover, the evaluation could be completed if authors could obtain an analysis of pre-implantation biopsies of the same patients already analysed, if possibile.
Undoubtedly, this analysis could improve our results. Unfortunately, we do not have enough preimplantation biopsies to perform it for two reasons; firstly, we do not perform preimplantation biopsies in all patients, and on the other hand, most of the grafts implanted in our program come from external centers, which makes it challenging to obtain these samples
Reviewer 2 Report
Insufficient experimental evidence for the authors claims, in addition to very low novelty. Insufficient English language and style. The authors only show one cell culture model as well as immunohistochemical stainings using human biopsy material. Data stemming from animal experiments (e.g. ischemia reperfusion model) would have been an obvious and necessary proof of the authors claims, but is lacking entirely.
Regarding the real time qPCR data shown, it appears that not all primer pairs are specific, including the housekeeping gene, thus making the data unreliable.
Beside the immunohistochemical data shown in figure 3 no data is shown on the protein expression level, and it appears that control biopsies also show extensive staining for C5aR1.
Author Response
REVIEWER 2
Insufficient experimental evidence for the authors claims, in addition to very low novelty. Insufficient English language and style.
We are grateful for your great review and contributions. We have done an extensive review and editing of our manuscript's grammar.
The authors only show one cell culture model as well as immunohistochemical stainings using human biopsy material.
Data stemming from animal experiments (e.g. ischemia reperfusion model) would have been an obvious and necessary proof of the authors claims, but is lacking entirely.
Our group has experience in renal ischemia-reperfusion injury on mice, and we have samples from these animals. However, after consideration, we decided this material would not contribute to shed light on the process because of the different pathways in complement activation described in animals.
Following the above, as we mentioned in our manuscript, multiple animal models have studied these receptors' expression; however, there is very little evidence from in vitro models with human cells or from the study of biopsies of kidney transplants. Therefore, we believe that our study provides additional data to those already available in the literature.
Regarding the real time qPCR data shown, it appears that not all primer pairs are specific, including the housekeeping gene, thus making the data unreliable.
We thank the reviewer for his/her comment. We used Primer3 and Primer-Blast-NCBI as tools for searching for specific primers, and later they were checked through Blast-NCBI for their specificity. Furthermore, after the Real-Time PCR program, LightCycler® 480 System adds an option to check the specificity of amplified cDNA fragments by the Melting curve analysis. Only one fragment is amplified and denatured at the same temperature when only one fluorescent peak is detected. When more than one amplicon is amplified in samples, more than one peak will be observed in the Melting curve.
Here we show Melting curves corresponding to some genes amplified in this study.
In the attached image, the first peak corresponds to KIM1, the first blue peak from the left corresponds to PPIA, and finally, the last blue peak corresponds to C5aR. Only one denaturing temperature recovers amplicon fluorescence indicating a specific amplification.
Beside the immunohistochemical data shown in figure 3 no data is shown on the protein expression level, and it appears that control biopsies also show extensive staining for C5aR1.
Thanks for the review and great comments. Unfortunately, we did not evaluate protein expression in our samples; however, we can affirm thatthe staining's specificity was corroborated by the meticulous evaluation from a pathologist with extensive experience in the study of kidney biopsies. Regarding the C5aR1 stain, the positive stain is not exclusive to biopsies with tubular damage. Similar to our results, a previous study performed in kidney transplant biopsies reported a positive stain in renal tubules from control biopsies (Ref: 23); however, both in that study and ours, the stain was predominantly observed in distal tubules. Our study detected diffuse staining of both proximal and distal tubules, especially in the biopsies from patients that experienced DGF. For this reason, we believe that this more diffuse expression could reflect a higher complement expression in the subset of the patients with DGF. We have tried to clarify this in the discussion section (lines 233-236)

Round 2
Reviewer 2 Report
Major:
The authors efforts to improve the manuscript are appreciated. The writing was improved, however, no new data was added and the existing data still lacks novelty.
As mentioned before the authors only use one cell culture model, in which they expose HK-2 cells to hypoxic conditions. The claim that, in doing so, they created an in vitro model of ischemia-reperfusion injury (IRI) is quite bold. HK-2 cells are an immortalized proximal tubule epithelial cell line from normal adult human kidney tissue and have been exposed to hypoxic conditions by other researchers before. This model can be useful to imitate the hypoxic aspect of ischemia reperfusion, however it is largely lacking other aspects involved in the development of IRI, especially in regard to the immune reaction.
Moreover, taking into consideration the authors findings in human biopsies that 1) C5aR1 was expressed in the apical and basal membrane of tubular epithelial cells, and 2) C5L2 deposits were observed in endothelial cells of peritubular capillaries) this cell culture model might not be suitable to investigate the latter and conclusions should be drawn carefully.
Animal data is still lacking, citing previous findings by other researchers is acceptable, underlines the lack of novelty however.
Although the manuscript has been edited, English language and style still need to be improved significantly, and new errors have been introduced in some places.
Author Response
The authors efforts to improve the manuscript are appreciated. The writing was improved, however, no new data was added and the existing data still lacks novelty.
As mentioned before the authors only use one cell culture model, in which they expose HK-2 cells to hypoxic conditions. The claim that, in doing so, they created an in vitro model of ischemia-reperfusion injury (IRI) is quite bold. HK-2 cells are an immortalized proximal tubule epithelial cell line from normal adult human kidney tissue and have been exposed to hypoxic conditions by other researchers before. This model can be useful to imitate the hypoxic aspect of ischemia reperfusion, however it is largely lacking other aspects involved in the development of IRI, especially in regard to the immune reaction.
R// Thanks for the time invested in reviewing our article and the recommendations.
Without any doubt, the ischemia-reperfusion process involves different and complex processes, in addition to hypoxia. Our model evaluated some components' dynamics from the complement system during the hypoxic-reoxygenation stimulus, mimicking some IRI events. So the best form to describe it is a hypoxia-reoxygenation model. We corrected the term throughout our manuscript.
Moreover, taking into consideration the authors findings in human biopsies that 1) C5aR1 was expressed in the apical and basal membrane of tubular epithelial cells, and 2) C5L2 deposits were observed in endothelial cells of peritubular capillaries) this cell culture model might not be suitable to investigate the latter and conclusions should be drawn carefully.
R// Thanks for the appreciations. We agree that our model cannot evaluate the interaction between C5L2 and the endothelial cells. For this reason, we admitted the necessity of additional studies, including hypoxic endothelial cell models, to evaluate the role of the C5L2 in endothelial hypoxic injury. We reflected that in the discussion.
However, we believe that the concordance of our results in the cellular model and histologic evaluation of kidney biopsies allows us to conclude that the C5L2 receptor plays an irrelevant role in the damage by complement system on tubular epithelial cells during hypoxia-reoxygenation or ischemia-reperfusion phenomenon.
Animal data is still lacking, citing previous findings by other researchers is acceptable, underlines the lack of novelty however.
R// We are grateful for the recommendation. Without a doubt, the animal models provide more information than cellular models; in our group, we have experience in murine models. However, some authors reported differences regarding complement functioning between murine and humans; this could restrict our results' extrapolation. For this reason, we focused on a human cellular model and kidney transplant biopsies. We added a sentence explaining this in the discussion section (lines 405-416)
Although the manuscript has been edited, English language and style still need to be improved significantly, and new errors have been introduced in some places.
R//We send our manuscript to MDPI English editing services. We added the English editing certificate as an additional document.